# Expression Patterns and Gonadotropin Regulation of the TGF-β II Receptor (Bmpr2) during Ovarian Development in the Ricefield Eel *Monopterus albus*

**DOI:** 10.3390/ijms232315349

**Published:** 2022-12-05

**Authors:** Zhi He, Li Zheng, Qiqi Chen, Sen Xiong, Zhide He, Jiaxiang Hu, Zhijun Ma, Qian Zhang, Jiayang He, Lijuan Ye, Liang He, Jie Luo, Xiaobin Gu, Mingwang Zhang, Ziting Tang, Yuanyuan Jiao, Yong Pu, Jinxin Xiong, Kuo Gao, Bolin Lai, Shiyong Yang, Deying Yang, Taiming Yan

**Affiliations:** 1College of Animal Science and Technology, Sichuan Agricultural University, Chengdu 611130, China; 2College of Veterinary Medicine, Sichuan Agricultural University, Chengdu 611130, China

**Keywords:** Bmpr2, immunoreactivity, FSH, hCG, *Monopterus albus*

## Abstract

Bmpr2 plays a central role in the regulation of reproductive development in mammals, but its role during ovarian development in fish is still unclear. To ascertain the function of *bmpr2* in ovarian development in the ricefield eel, we isolated and characterized the *bmpr2* cDNA sequence; the localization of Bmpr2 protein was determined by immunohistochemical staining; and the expression patterns of *bmpr2* in ovarian tissue incubated with FSH and hCG in vitro were analyzed. The full-length *bmpr2* cDNA was 3311 bp, with 1061 amino acids encoded. Compared to other tissues, *bmpr2* was abundantly expressed in the ovary and highly expressed in the early yolk accumulation (EV) stages of the ovary. In addition, a positive signal for Bmpr2 was detected in the cytoplasm of oocytes in primary growth (PG) and EV stages. In vitro, the expression level of *gdf9,* the ligand of *bmpr2*, in the 10 ng/mL FSH treatment group was significantly higher after incubation for 4 h than after incubation for different durations. However, *bmpr2* expression in the 10 ng/mL FSH treatment group at 2 h, 4 h and 10 h was significantly lower. Importantly, the expression level of *bmpr2* and *gdf9* in the 100 IU/mL hCG group had similar changes that were significantly decreased at 4 h and 10 h. In summary, Bmpr2 might play a pivotal role in ovarian growth in the ricefield eel, and these results provide a better understanding of the function of *bmpr2* in ovarian development and the basic data for further exploration of the regulatory mechanism of *gdf9* in oocyte development.

## 1. Introduction

TGF-β family members participate in ovarian development in female animals and are related to reproductive regulation [1,2]. Their effects on target cells work by binding to type I and type II receptors containing serine/threonine kinase and the formation of heteromeric complexes [3,4,5]. In mammals, Bmpr2 is a type II receptor of the TGF-β family that can respond to TGF-β signaling and precipitate a cascade reaction involving downstream genes [6]. BMPR2 is a key factor involved in regulating signal transduction during gonadal development in animals. *Bmpr2* mRNA transcripts were previously detected in human ovaries, specifically in granulosa cells (GCs) [7,8]. Similarly, Bmpr2 mRNA expression has been detected in sheep GCs and oocytes [9]. Furthermore, immunoblotting analysis has shown that BMPR2 is expressed in GCs and follicles in water buffalo [10], and other studies have reported that Bmpr2 mRNA is expressed in the different stages of porcine follicular development [11]. The Bmpr2 mRNA is expressed obviously in the cumulus cells and oocytes of pigs during in vitro maturation [12,13]. Numerous research results have shown that BMPR2 plays a major role in the growth and functioning of follicles in mammals and chickens [14]. However, there are few studies on *bmpr2* in fish. Compared with other tissues, the *bmpr2* mRNA expression levels were significantly higher in the ovary in *Schizothorax prenanti*. More importantly, *bmpr2* expression continued to increase to its peak in the ovary in the cortical alveoli stage (CAS) and then subsequently decreased in *S. prenanti* [15]. The expression levels of type II Bmp receptors (*bmpr2a* and *bmpr2b*) in the zebrafish ovary consistently increased during folliculogenesis, with the peak levels occurring in the full-growth stage prior to final oocyte maturation [16]. These results suggest that *bmpr2* is also involved in reproductive development in fish.

Ricefield eel (*Monopterus albus)* is a significantly cultured fish in China and an essential animal protein source for people. The artificial reproduction of the ricefield eel is seriously hindered by the small size of the female, the small number of eggs and low fecundity [17]. Therefore, it is very important to explore the regulatory mechanism of embryonic ovarian development. Our previous report showed that *gdf9* was specifically expressed in the ovaries of the ricefield eel and was closely related to early folliculogenesis [18]. However, how the signal of GDF9 is transmitted to the target cell remains unclear. Therefore, we aimed to further explore Bmpr2, the receptor of GDF9 [18]. Towards a better understanding of the role of *bmpr2* on ovarian development in the ricefield eel, the *bmpr2* cDNA sequences were cloned, the expression of *bmpr2* mRNA levels was evaluated, and the localization of Bmpr2 in ovarian tissues of the ricefield eel was detected. Furthermore, the expression levels of *bmpr2* and *gdf9* based on the incubation with human chorionic gonadotropin (hCG) and follicle-stimulating hormone (FSH) in fresh ovarian fragments were detected in vitro. FSH and hCG are gonadotropins that can regulate ovarian development [19], and they are used for the artificial maturation of many fishes [20,21,22]. These results provide a better understanding the functions of *bmpr2* and the regulatory mechanism of *gdf9* in ovarian development and help to provide the basic data for the research of artificial propagation technology of the ricefield eel.

## 2. Results

### 2.1. Cloning Sequence Analysis of Ricefield Eel Bmpr2

The full-length sequence of ricefield eel *bmpr2* was 3311 bp comprising a 62-bp 5′UTR, an ORF of 3186 bp and a 63-bp 3′UTR. The ORF of *bmpr2* encodes 1061 amino acids. Four N-glycosylation sites were predicted by Expasy Prosite, namely, N^64^, N^120^, N^136^ and N^803^ (Figure 1). The theoretical Mw and pI of Bmpr2 are 115.38 kDa and 5.70, respectively. Bmpr2 is hydropathic, and its grand average hydropathicity is −0.531. The phylogenetic tree was constructed based on the full protein sequences of BMPR2 orthologues (Figure 2A). The Bmpr2 of the ricefield eel clustered with several perciform fishes by phylogenetic analysis. Overall, Bmpr2 of the ricefield eel was conservative in the ligand-binding domain (ActRI/ActRII domain) and kinase domain. Compared with the ligand domain and carboxy-terminal tail, the kinase domain was more conservative in animal Bmpr2 (Figure 2B). The Bmpr2 kinase domain of the ricefield eel was more conserved compared to zebrafish Bmpr2b (85.57%) and zebrafish Bmpr2a (78.86%), and the kinase domain sequence consistency was between 95.64% (*Scatophagus argus* Bmpr2b) and 96.31% (*Micropterus salmoides* Bmpr2b) (Figure 2B). The results of the phylogenetic tree and Bmpr2 sequence structure characteristics showed that ricefield eel type II Bmp receptor had higher similarity with Bmpr2b of bony fishes. However, according to genome analysis (gene: ENSMALG00000012847), there was only one type of *M. albus* Bmpr2. Taken together, the sequence we cloned was finally defined as Bmpr2.

### 2.2. Tissue Distribution of bmpr2 mRNA

*bmpr2* mRNA expression was detected in various tissues, such as the brain, heart, liver, kidneys, intestines, spleen, blood, ovaries, testes, and intersex gonads. The expression levels of *bmpr2* mRNA in the ovary were significantly higher than those in other tissues (Figure 3A). The size of Bmpr2 from the ovary of *M. albus* was in accord with the theoretical molecular weight by Western blot analysis (Figure 3B).

### 2.3. Localization and Expression Levels of Bmpr2 in Developing Ovaries

As shown in Figure 4A, compared with the primary growth stage ovary (PG, containing oocytes without cortical alveoli), *bmpr2* mRNA expression was obviously increased in the previtellogenic stage of the ovary (PV, containing oocytes with the appearance of cortical alveoli) and the early vitellogenin stage of the ovary (EV, containing oocytes at early vitellogenesis) and reached a maximum in the EV stage. However, it decreased significantly in the middle to late vitellogenic stage of the ovary (MLV, containing oocytes with active vitellogenesis) and the oocyte in the mature stage of the ovary (OM, containing mature oocytes) (Figure 4A).

Furthermore, Bmpr2 signals were detected in the PG stage and EV stage by immunohistochemical analysis. In the primary growth (PG) stage, the Bmpr2-positive signal was detected in the cytoplasm of oocytes (Figure 4B(i)). In the early vitellogenin (EV) stage, the Bmpr2-positive signal was located at the cytoplasm of the early-vitellogenic stage oocytes and follicle cells (Figure 4B(ii)).

### 2.4. Colocalization of Ricefield Eel Bmpr2 and Gdf9 in Ovaries

We detected the immunofluorescence signals of Bmpr2 and Gdf9 in the three stages (PG, PV and EV, Figure 5F,f). As shown in Figure 5, we can see the primary growth stage oocyte, pre-vitellogenic stage oocyte, and early vitellogenic-stage oocytes. The fluorescence signal of Bmpr2 (FITC, green signal, Figure 5B,b) was detected in the cytoplasm of oocytes. The Bmpr2 signal in PG stage oocytes was stronger than that in PV stage and EV stage oocytes (Figure 5b). The localization results of Gdf9 (TRITC, orange-red signal, Figure 5D,d) in *M. albus* ovaries were similar. The fluorescence signal of Bmpr2 and Gdf9 in early oocytes was stronger, and there was a signal in the cytoplasm of oocytes.

### 2.5. Expression of bmpr2 and gdf9 in Ovarian Tissue of Ricefield Eel after FSH and hCG Incubation In Vitro

The expression level of *gdf9* was significantly enhanced and maintained at a high level after stimulation with different concentrations of FSH at 4 h (Figure 6A). Subsequently, the expression of *gdf9* in the 0.5 ng/ mL and 1 ng/mL FSH treatment groups at 10 h was significantly decreased (Figure 6A). The expression of *bmpr2* was inhibited with increasing treatment time after stimulation with different concentrations of FSH, and *bmpr2* expression was continuously inhibited in the high-dose group (FSH 10.0 ng/mL) 1 h later (Figure 6C).

The expression levels of *gdf9* and *bmpr2* in the 10 IU/mL hCG group did not significantly change with the increase in incubation time (Figure 6B,D). There was a tendency for the expression levels of *gdf9* and *bmpr2* in the 100 IU/mL hCG group to increase with the increase in incubation time, which reached the peak at 2 h (Figure 6B,D). Subsequently, *gdf9* expression in the 100 IU/mL hCG group was significantly decreased at 4 h and 10 h (*p* < 0. 0001). Similarly, *bmpr2* expression showed the same changes in the 100 IU/mL hCG group at 4 h and 10 h (Figure 6D).

## 3. Discussion

In this study, we cloned 3311 bp *bmpr2* cDNA sequences from the gonads of *M. albus* and characterized them. Type II TGFβRs classical structural features contain a ligand binding domain, a single transmembrane domain, an intracellular serine/threonine kinase domain and a carboxy-terminal tail [23,24]. Despite the sequence divergence, the results showed that the ligand binding domain (ActRI/ActRII domain) and kinase domain of *M. albus* sequence were highly conserved with other animals, including species such as *Homo sapiens, Gallus, M. salmoides* and *S. argus*. The ActRI/ActRII domain and kinase domain play a crucial role in ligand recognition and activation, similar to the results surveyed in *Anguilla australis* [25] and *S. prenanti* [15]. The results of the phylogenetic tree and Bmpr2 sequence structure characteristics showed that ricefield eel type II Bmp receptor had higher similarity with Bmpr2 of bony fishes, especially with Bmpr2b. However, unlike zebrafish [26], according to genome analysis (gene: ENSMALG00000012847, http://www.ensembl.org/index.html, accessed on 2 December 2022), there was only one type of *M. albus* Bmpr2 on the chromosome. In summary, we cloned the sequence, which was finally defined as Bmpr2.

Although *bmpr2* mRNA is widely expressed in animal tissues, the tissues that can highly express *bmpr2* mRNA are different in various species. For example, in the short-finned eel, *bmpr2* mRNA is expressed in the muscle, heart, eye, liver, hindbrain and ovary, among other tissues, but it is expressed mainly in the thyroid [25]. In zebrafish, *bmpr2a* and *bmpr2b* mRNA expression occurs mainly in the gill and testis [16]. In this paper, we found that the tissue expression patterns of *bmpr2* mRNA in ricefield eels were markedly higher in the ovary than in other tissues, consistent with those of *bmpr2* in *S. prenanti* [15].

Although *bmpr2* is not specific to ovarian tissues, it plays a pivotal role during ovarian development. Over the course of the ovarian cycle in rats, BMPR-II mRNA expression in the GCs of primordial follicles was not detected, was low in primary follicles, and increased to high maximal levels in the GCs of secondary follicles [27]. Strong immunostaining for BMPR2 in sheep ovaries has been found in the primary to late antral stages of ovarian development [28]. In this study, we showed that *bmpr2* mRNA expression in the ovaries of ricefield eels reached its maximum in the EV stage. The result was similar to the *bmpr2* mRNA expression pattern in short finned eel [16]. In order to make further efforts to understand the impact of *bmpr2* during ovarian development, we evaluated the localization of Bmpr2 protein by immunohistochemical staining. Strong immunostaining for Bmpr2 was observed in the cytoplasm of oocytes in the PG stage and EV stage in ricefield eels. BMP receptors are expressed in pigs in different developmental stages, and positive immunostaining of Bmpr2 protein was observed in the GCs and theca cells of prepubertal ovaries [11]. In mouse follicles, similar to porcine follicles, oocytes and GCs of the primordia showed strong positive staining for BMPR2 [29]. The expression of BMPR2 mRNA in the theca cells and granulosa cells of bovine ovarian follicles has also been confirmed [30]. These studies suggest that the expression patterns of Bmpr2 vary among species.

The secretion of paracrine factors by oocytes plays a central role in the early follicular growth and development stage, including from the primordial to primary follicles [31,32,33,34]. GDF9 is an important paracrine factor secreted by oocytes [35], and in 1995, it was demonstrated to be oocyte specific [36]. Gdf9 mRNA expression has been found in oocytes in bovines [37], rabbits [38], sheep [39], chicken [40], zebrafish [41] and ricefield eel [18] in the early stages of follicular development. Female mice lacking the GDF9 gene can still form primordial follicles, but the oocytes stop growing when they reach the primary follicle stage, leading to complete infertility [42]. These studies suggested that *gdf9* may be involved in early folliculogenesis. Our study found that Bmpr2 and Gdf9 were colocalized to oocyte cytoplasm in the three stages (PG, PV and EV), and the signal in PG stage oocytes was stronger than that in PV stage and EV stage oocytes. The mRNAs for GDF9 and BMPR2 were detected in primordial and secondary follicles as well as in the oocyte and GCs of the antral follicles [43]. These findings show that *gdf9* and *bmpr2* are essential for ovarian growth.

Vertebrate reproduction is controlled by gonadotrophic hormones [44,45]. In the follicle, FSH and hCG influence multiple intracellular networks [46,47,48] and have a positive effect on oocyte maturation [49]. GDF9 and FSH stimulate preantral follicle growth in rats, and GDF9 significantly enhances follicular growth induced by FSH [50]. FSH induces preantral follicular growth through the upregulation of FSHR, a mechanism mediated by the expression and action of oocyte-derived GDF9 [51]. In addition, GDF9 contributes to FSHβ expression both in immortalized gonadotropic cells and in primary pituitary cultures and regulates FSHβ expression [52]. Intriguingly, FSH in turn affects the expression and function of both GDF9 and its receptors. Evidence shows that the expression of GDF-9 is increased by FSH and might be involved in FSH stimulated follicular development [53]. This study showed that the expression level of *gdf9* was significantly enhanced and maintained at a high level after stimulation with different concentrations of FSH (0.5 ng/mL, 1 ng/mL and 10 ng/mL) at 4 h. Thus, this indicates the effect of FSH on Gdf9, possibly through the “Gdf9-FSH loop” on inducing ovarian development. Strangely, the effect FSH on GDF9 receptors is different. The expression of BMPRII is downregulated by treatment with FSH alone in cultured bovine GCs [30]. At present, our study found that FSH downregulated the expression of the *gdf9* receptor gene *bmpr2*. The expression of *bmpr2* was inhibited with increasing treatment time after stimulation with different concentrations of FSH (0.5 ng/mL, 1 ng/mL and 10 ng/mL), and *bmpr2* expression was continuously inhibited in the high-dose group (FSH 10.0 ng/mL) 1 h later. A previous study showed that in cultured bovine GCs treated with different concentrations of FSH (0 ng/mL, 1 ng/mL, 5 ng/mL and 10 ng/mL), and FSH also downregulated the expression of Bmpr2 [54]. However, adding 5 ng/mL of FSH to cultured sheep GCs in vitro had no significant effect on Bmpr2 mRNA expression, while adding 1 ng/mL or 10 ng/mL of FSH downregulated Bmpr2 mRNA expression [55]. Another study showed that estrogen suppresses Bmpr2 expression through direct Bmpr2 promoter binding by the estrogen receptor [56]. Bmpr2 inhibition may be due to the binding of the estrogen receptor to its promoter. These findings suggest that the effect of FSH concentration on *bmpr2* mRNA expression may vary from species to species. However, the inhibitory effect of FSH on Bmpr2 remains to be further studied.

hCG can interact with the fish LH receptor [57] and regulate steroid production and oocyte maturation [58,59]. In this study, we found that the expression levels of *gdf9* and *bmpr2* in the 100 IU/mL hCG group were increased with the increase in incubation time and reached the peak at 2 h. Subsequently, *gdf9* and *bmpr2* expression in the 100 IU/mL hCG group was significantly decreased. Interestingly, treatment of isolated ovaries from zebrafish with different concentrations of hCG (0 IU/mL, 1 IU/mL, 10 IU/mL and 100 IU/mL) for 2 h successively reduces the expression of *gdf9* mRNA, and *gdf9* is significantly suppressed by hCG in high dose (100 IU/mL) groups [41]. In response to hCG treatment of the flounder ovary cell line, 4 h after treatment, the expression of *gdf9* is gradually upregulated at hCG concentrations of 1 IU/mL and 10 IU/mL and significantly downregulated at concentrations of 100 IU/mL and 1000 IU/mL [60]. These results suggest that there may be a species-specific negative feedback loop in *gdf9* regulated sex hormone biosynthesis. When the concentration of sex hormones is high, *gdf9* downregulation is beneficial for regulating hormone biosynthesis processes in the ovary. Furthermore, BMPR2 is a crucial receptor for GDF9. The effects of GDF9 might be transduced by binding to BMPR2 [61]. Evidence shows that hCG (200 IU/mL) can downregulate Bmpr2 mRNA expression in vitro cultured human luteinized granulosa cells [62]. This information indicates that hCG is a major ovarian signaling factor that plays an important role in ovarian development. Whether *gdf9* and *bmpr2* levels regulate follicular development and oocyte maturation and whether these two molecules cooperate to control folliculogenesis in fish remain to be determined.

## 4. Materials and Methods

### 4.1. Experimental Fish

Ricefield eels (*n* = 200, body weight = 65.2 ± 45.9 g and body length = 41.26 ± 7.13 cm) were purchased from a local market (Chengdu, Sichuan, China). Ricefield eels were cultivated in natural conditions of a temperature of 21.7 ± 2.5 °C and photoperiod of 16 h light:8 h dark. The aquaculture system’s water environment included dissolved oxygen (more than 6 mg/L), pH (6.5–7.4) and ammonia nitrogen (<0.2 mg/L). The Animal Research and Ethics Committees of Sichuan Agricultural University reviewed and approved all procedures and investigations. All procedures and investigations were conducted in accordance with the guidelines of the committee (approval no: 20170031). Fish were decapitated after anesthetized with MS-222 (100 μg/mL, Syndel, WA, USA) for 10 min. The gonads, pituitary, heart, liver, kidney, intestine, spleen, blood and other tissues of ricefield eels were collected and then stored at −80 °C for RNA extraction after quickly freezing with liquid nitrogen.

According to the method of He et al. [18], the other parts of gonads were used for histological observation after a series of dehydration, embedding, sectioning (5 µm sections) and staining with hematoxylin–eosin. The gonads were classified as follows: the primary-growth-stage ovary (PG, containing oocytes without cortical alveoli), the previtellogenic stage ovary (PV, containing oocytes with the appearance of cortical alveoli), and the early vitellogenin stage ovary (EV, containing oocytes at early vitellogenesis), the middle to late vitellogenic stage ovary (MLV, containing oocytes with active vitellogenesis) and the oocyte at mature stage ovary (OM, containing mature oocytes).

### 4.2. Isolation of Total RNA and Transcribed into cDNA

Total RNA was isolated from tissues stored in liquid nitrogen of ricefield eels using TRIzol reagent (Invitrogen, Chicago, IL, USA). Total RNA isolated from tissues was first treated by DNase I (Thermo Scientific, Waltham, MA, USA) to remove genomic or avoid DNA contamination. The quality and integrity of the RNA were assessed by the absorbance at 260 and 280 nm and 1% agarose gel electrophoresis. The RNA of ovaries was transcribed into cDNA using RT Reagent Kit (Thermo Scientific, Waltham, MA, USA). The specific operation was carried out according to its instructions.

### 4.3. Gene Cloning and RT-qPCR

The *bmpr2* cDNA was cloned, and its specific primers are shown in Table 1. *bmpr2* cDNA was designed according to the sequence of ricefield eels *bmpr2* in the NCBI database (ID: 109951513). Ovary cDNA was used as the template, and specific primers and Taq DNA Buffer (Thermo Scientific, Waltham, MA, USA) were added to PCR tubes for reaction. The amplification program was as follows: initial 3 min denaturation at 95 °C; 35 cycles of 95 °C for 0.5 min, annealing of 54~55 °C for 0.5 min and extension of 72 °C for 1.5 min; and final extension of 72 °C for 20 min. All amplified fragments were ligated into pMD19-T (Takara, Dalian, China). The positive cloning vector was checked by Sangon Biological (Shanghai, China).

The *bmpr*2 mRNA specific primers (Table 1) were designed according to the ricefield eel *bmpr2* sequence in the NCBI database (ID: 109951513). *ef1α* and *rpl17* were most stable in different tissues of *M. albus* [63]. Therefore, *ef1α* and *rpl17* were used as internal controls. The expression level of *bmpr2* in the tissues and developing ovaries of ricefield eel was analyzed by RT-qPCR. The cDNA of various tissues and developing ovaries in the ricefield eel was used as the template. The RT-qPCR mixture contained 1 μL of cDNA template, 5 μL of 2× SYBR Green MasterMix (TaKaRa Bio, Dalian, China), 0.4 μL upstream/downstream primers and 3.2 μL of ddH_2_O. RT-qPCR was carried out at 95 °C for 3 min, followed by 40 cycles of 95 °C for 10 s, 57 °C for 30 s and then by signal collection. The melting curve was generated as follows: 95 °C for 5 s, followed by heating from 65 °C–95 °C while increasing the temperature by 5 °C every 5 s and then by signal collection. After the reaction was completed, IQ5 analysis software was used for automatic results. The PCR products were assessed by 2% agarose gel electrophoresis. The practical methods refer to previously published articles [64].

### 4.4. Analysis of bmpr2 Sequence Characteristics

The open reading frame (ORF) of *bmpr2* was predicted using the web-based tool ORF Finder (https://www.ncbi.nlm.nih.gov/orffinder/ (accessed on 5 June 2022), and the DNA sequence was spliced and translated by the DNAMAN software translation tool (version 6.0, Lynnon Biosoft, San Ramon, CA, USA). The molecular weight (Mw) and theoretical isoelectric point (pI) were searched by online software (https://web.expasy.org/compute_pi/ (accessed on 9 June 2022). Expasy Prosite (http://www.cbs.dtu.dk/services/NetNGlyc/ (accessed on 9 June 2022) was used to determine N-glycosylation sites. The TMHMM web server v2.0 (http://www.cbs.dtu.dk/services/TMHMM/ (accessed on 9 June 2022) and SMART version 9 (http://smart.embl-heidelberg.de/ (accessed on 9 June 2022) [65] were used to predict the protein domains. The phylogenetic analyses were constructed by the neighbor-joining method with bootstrap values calculated from 1000 replicates in MEGA (version 11.0, Mega Limited, Auckland, New Zealand) [66].

### 4.5. Western Blot Analysis

BMPR2 (LifeSpan BioSciences, Seattle, WA, USA; cat. no. LS-C178875) was purchased as the primary antibody. BMPR2 (LS-C178875) antigen sites are located at 295–552aa. Before purchasing the primary antibody, the Bmpr2 sequence alignment between *M. albus* and *H. sapiens* BMPR2 (LS-C178875) was carried out using the DNAMAN software translation tool (version 6.0, Lynnon Biosoft, San Ramon, CA, USA). The homology of these two sequences was 82.17% (Appendix A). Horseradish peroxidase (HRP)-conjugated goat anti-rabbit IgG (Boster, Wuhan, China) was used as a secondary antibody. Western blot analysis was conducted as described in our earlier study [18]. Briefly, ovary protein was extracted with RIPA Lysis Buffer and 1 Mm PMSF (Beyotime Biotechnology, Shanghai, China). The homogenate was centrifuged at 12,000× *g* for 3 min at 4 °C, and then the protein concentration was quantitated by a BCA kit (Beyotime Biotechnology, Shanghai, China). Equal proteins were added to each lane of 10% SDS-PAGE gel for protein separation and then transferred to PVDF membranes (Merck Millipore, Darmstadt, Germany) by the Bio-Rad Trans-Blot system. After blocking the membrane with 5% non-fat milk for 4 h at room temperature, the membrane was incubated with the primary antibody (final dilution, 1:800) for 2 h at room temperature. The membrane was rinsed with 10 mM of PBST 3 times for 5 min each. Then, the membrane was hatched using HRP-labeled goat anti-rabbit IgG (Boster, Wuhan, China). The protein bands were evaluated by Bio-Rad ChemiDocTM MP (Bio-Rad, Hercules, CA, USA) and checked by the DAB kit (Maxim, Fuzhou, China). The primary antibody was replaced with PBS in the negative control group (Appendix A).

### 4.6. Immunohistochemistry Analysis

Paraffin sections (4–6 μm) of the ovarian tissues were incubated in 3% H_2_O_2_ for 0.5 h to remove endogenous peroxidase activity. The sections of the ovarian tissues were washed with phosphoric acid buffer (PBS, 10 mM) and incubated with 10% goat serum (Boster, Wuhan, China) for 20 min. Thence, the sections were incubated with rabbit BMPR2 antibody serum (diluted at 1:300) at 25 °C for 2 h. The sections were then washed with PBS and incubated with HRP-conjugated goat anti-rabbit IgG for 30 min. After washing with PBS, the sections were detected with an ECL reagent kit and observed with a microscope digital imaging system (Nikon, Tokyo, Japan). The primary antibody was replaced with PBS as the negative control group and used to confirm the specificity of the immunostaining.

### 4.7. Immunohistochemical Colocalization Analysis

Frozen sections of ricefield eel ovaries were taken for Bmpr2 and GDF9 fluorescence immunohistochemical localization. Gdf9 polyclonal antibody was generated as previously described [18]. The frozen slices were removed from the refrigerator at −20 °C and dried at room temperature. The ovarian tissues were fixed with 4% paraformaldehyde for 10 min and then rinsed with PBS buffer. PBS solution was discarded, and 2~3 drops of normal goat serum (Boster, Wuhan, China) were added to each section at room temperature and placed in a wet box for incubation for 20 min. Then, 50 μL of Bmpr2 or GDF9 rabbit antiserum (diluted at 1:300) was added dropwise to each section, and the sections were placed in a wet box for overnight incubation at 4 °C. Then, the sections were washed with PBS buffer, the PBS solution was discarded, 100 μL of FITC-Sheep anti-rabbit IgG (Boster, Wuhan, China) (1:500 dilution) was added to each section, and the sections were incubated in a wet box away from light at room temperature for 60 min. Afterwards, the sections were washed with PBS, the PBS solution was discarded, and 100 μL of DAPI was added to each section to stain the nuclei. The samples were incubated at room temperature in a wet box away from light for 0.25 h and washed with PBS. Two drops of anti-fluorescence quenching sealing tablets were added to each section, and the tablets were sealed. The sections were then stored away from light, and photographs were taken using a forward fluorescence microscope. The negative control group of immunofluorescences had no signal, and there is no display here.

### 4.8. Effect of hCG and FSH on bmpr2 and gdf9 Expression

In vitro, freshly ovarian fragments (50–100 mg) were placed in 24-well tissue culture dishes in 1 mL of Leibovitz L-15 medium (Gibco, Shanghai, China) containing penicillin (0.1 U/mL, Gibco, Shanghai, China) and streptomycin (0.1 mg/mL, Gibco, Shanghai, China) for preincubation for 2 h at 28 °C. After the ovarian fragments were treated with L-15 medium containing hCG (Sigma-Aldrich Trading Co. Ltd., Shanghai, China, Cas no. 9002-61-3) at 10, 50 and 100 IU/mL, FSH (Sigma-Aldrich Trading Co., Ltd., Shanghai, China, Cas. no. 9002-68-0) at 0.5, 1 and 10 ng/mL or normal saline (control group) for 1 h, 2 h, 4 h or 10 h. After incubation, total RNA was isolated from ovarian fragments and transcribed into cDNA using RT Reagent Kit (Thermo Scientific, Waltham, MA, USA) for the analysis of *bmpr2* and *gdf9* expression by RT-qPCR above. The *bmpr2*-mRNA (ID: 109951513) and *gdf9* mRNA (ID:109971157) specific primers (Table 1) were designed according to the ricefield eel *bmpr2* sequence in the NCBI database. Results are expressed as means ± SEM (error bars), which were compared with the control group at the same treatment time. The specific methods refer to previously published articles [64].

### 4.9. Statistical Analysis

The results were presented as means ± SEM. Statistical analysis was conducted using SPSS 21.0 software (SPSS, Inc., Chicago, IL, USA). The data were analyzed using one-way (ANOVA) and two one-way ANOVA followed by Duncan’s multiple comparisons test using SPSS 21.0 (IBM, Armonk, NY, USA). Finally, Pearson’s product moment was used for correlation analysis. *p* < 0.05 was considered statistically significant.

## 5. Conclusions

In summary, *bmpr2* was isolated and characterized from the gonads of the ricefield eel. *bmpr2* mRNA was substantially expressed in the EV phase during ovarian development. Bmpr2 immunoreactive signals were observed in the cytoplasm of oocytes. In vitro, FSH and hCG regulate the expression of *bmpr2* and *gdf9* in ovary. These findings show that *gdf9* and *bmpr2* might partake in folliculogenesis in the ricefield eel. Our results showed that FSH and hCG can affect the expression of *gdf9* and *bmpr2* in the ovary of ricefield eel; however, the regulatory mechanism remains to be explored. Further research is required to investigate whether FSH and hCG help in artificial breeding.

## Figures and Tables

**Figure 1 ijms-23-15349-f001:**
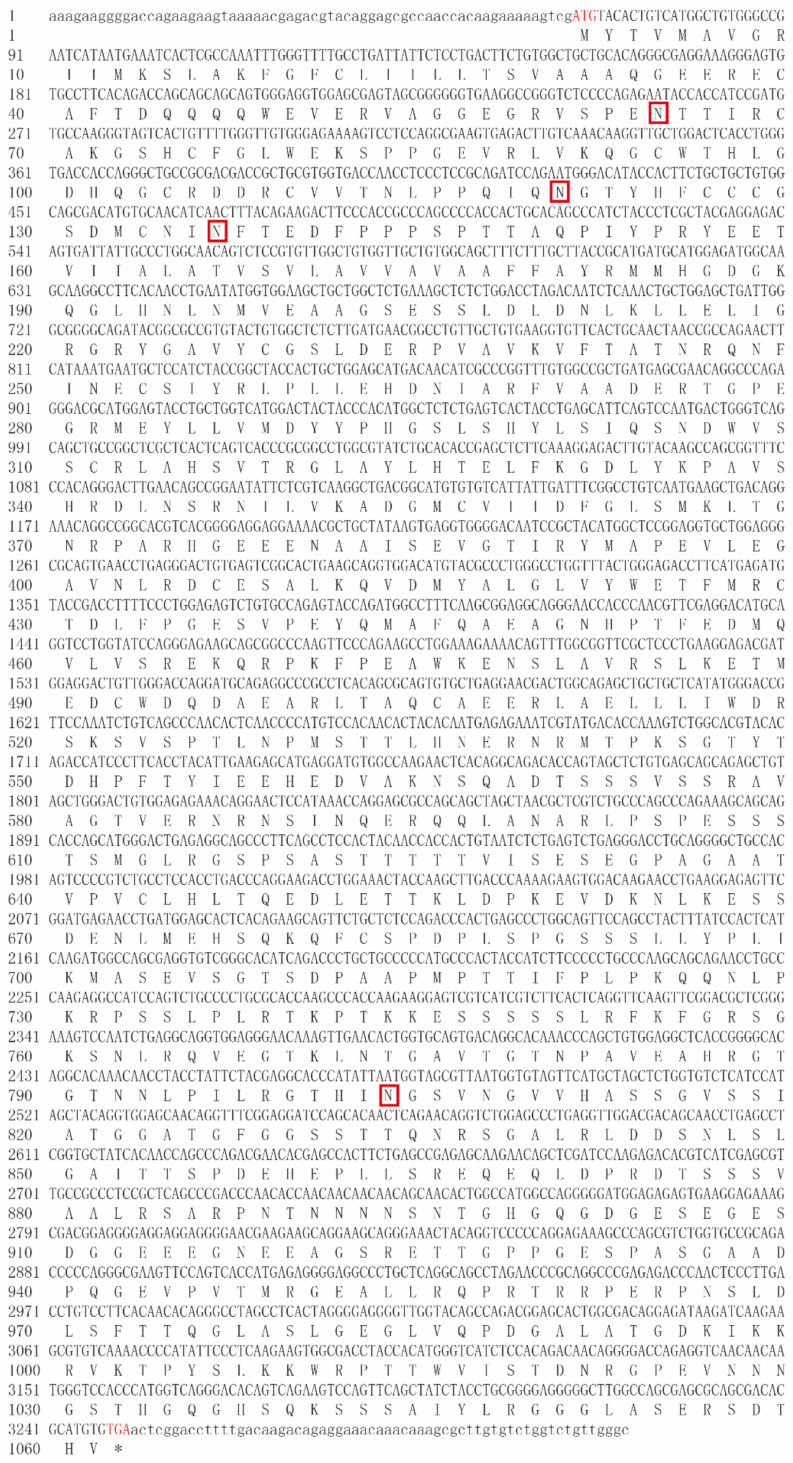
Nucleotide sequence information and amino acid sequence of the coding region of Bmpr2 in ricefield eel, *Monopterus albus*. The untranslated regions and translated regions are indicated by lowercase letters and uppercase letters, respectively. The predicted N-glycosylation sites are red boxes. The initiation codon (ATG) and stop codon (TAA) are marked in red color. Asterisks (*) indicate the translation stop codon.

**Figure 2 ijms-23-15349-f002:**
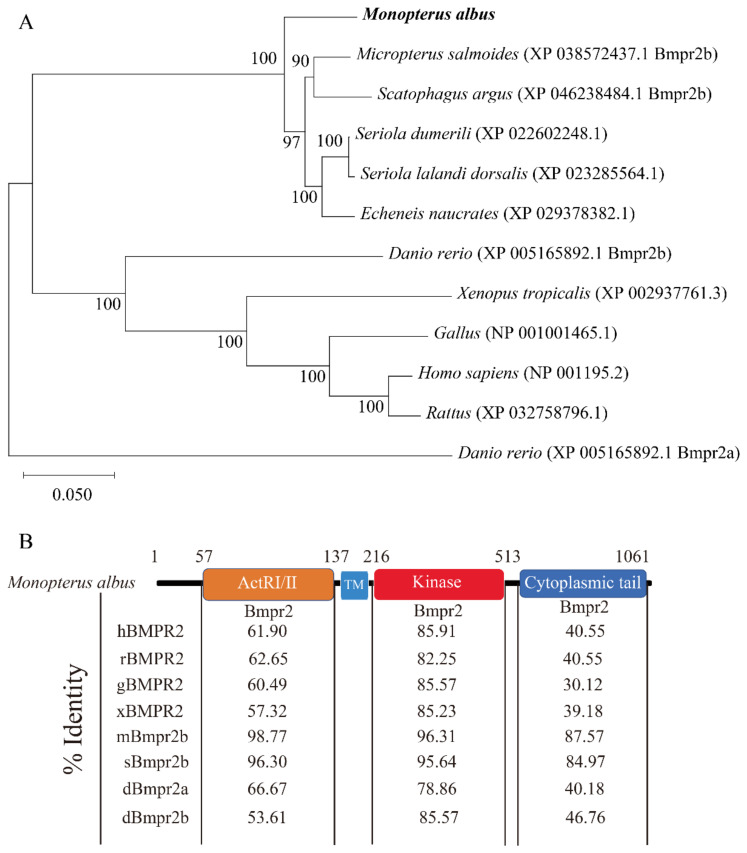
Phylogenetic analysis and domain characteristics of Bmpr2 in ricefield eel, *Monopterus albus*. (**A**), The phylogenetic tree was researched by the neighbor-joining algorithm of Mega 11. The phylogeny was tested using the bootstrap method with 1000 replications. The numbers at nodes are bootstrap values (%). *M. albus* is marked in bold. The Bmpr2 protein sequences of vertebrates were obtained from Entrez (NCBI). (**B**), The characteristic Bmpr2 domains are conserved in ricefield eel orthologues. Bmpr2 is conservative in ligand-binding domain (ActRI/ActRII domain) and kinase domain, while the kinase domain is more conserved in Bmpr2. Numbers represent percentage identity of the predicted protein sequences with other type II BMP receptor orthologues (hBMPR2—*Homo sapiens* BMPR2, rBMPR2—*Rattus* BMPR2, gBMPR2—*Gallus* BMPR2, xBMPR2—*Xenopus tropicalis* BMPR2, mBmpr2—*Micropterus salmoides* Bmpr2b, sBmpr2—*Scatophagus argus* Bmpr2b, dBmpr2a—*Danio rerio* Bmpr2a, and dBmpr2b—*Danio rerio* Bmpr2b).

**Figure 3 ijms-23-15349-f003:**
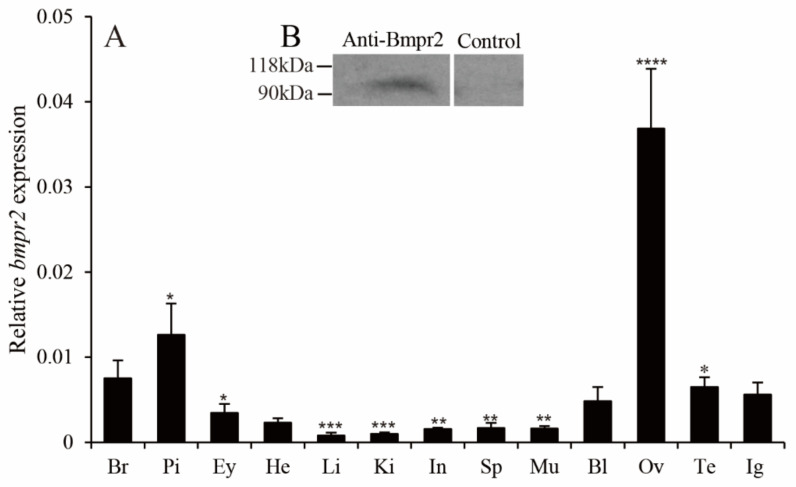
Expression of *bmpr2* in different tissues of ricefield eel, *Monopterus albus*. (**A**), Relative mRNA levels of *bmpr2* in tissues of ricefield eels were analyzed using RT-qPCR. (**B**), Bmpr2 immunoreactivity in the ovaries was evaluated using Western blot analysis. Anti-Bmpr2, primary antibody; Control, negative control; Bl, blood; Br, brain; Ey, eyes; He, heart; Ig, intersex gonads; In, intestines; Ki, kidneys; Li, liver; Mu, muscle; Ov, ovaries; Pi, pituitary; Sp, spleen; Te, testes. Results are expressed as means ± SEMs (*n* = 5). *, **, *** and **** are the significant differences of Duncan’s multiple comparisons, representing * *p* < 0.05, ** *p* < 0.01, *** *p* < 0.001 and **** *p* < 0.0001, respectively.

**Figure 4 ijms-23-15349-f004:**
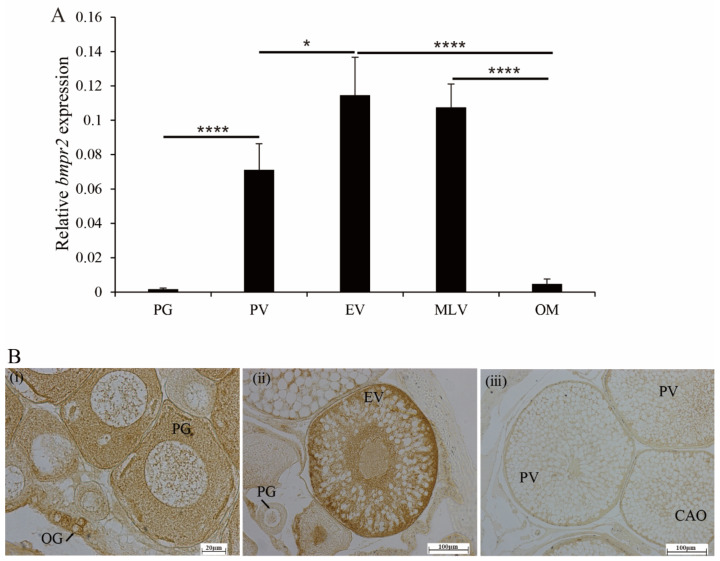
Expression of *bmpr2* in developing ovaries was analyzed in ricefield eel, *Monopterus albus*. (**A**), Expression of *bmpr2* during ovarian development. (**B**), The localization of Bmpr2 immunoreactive signals (yellow and dot-like) in the ovary. (**i**), Primary growth stage; (**ii**), early vitellogenic stage; (**iii**), negative control. EV, early vitellogenic stage; MLV, middle to late vitellogenic stage; OM, oocyte in mature stage; OG, oogonium; PG, primary growth stage; PV, previtellogenic stage. Results are expressed as means ± SEMs (*n* = 5). * and **** are the significant differences in Duncan’s multiple comparisons, representing * *p* < 0.05 and **** *p* < 0.0001, respectively.

**Figure 5 ijms-23-15349-f005:**
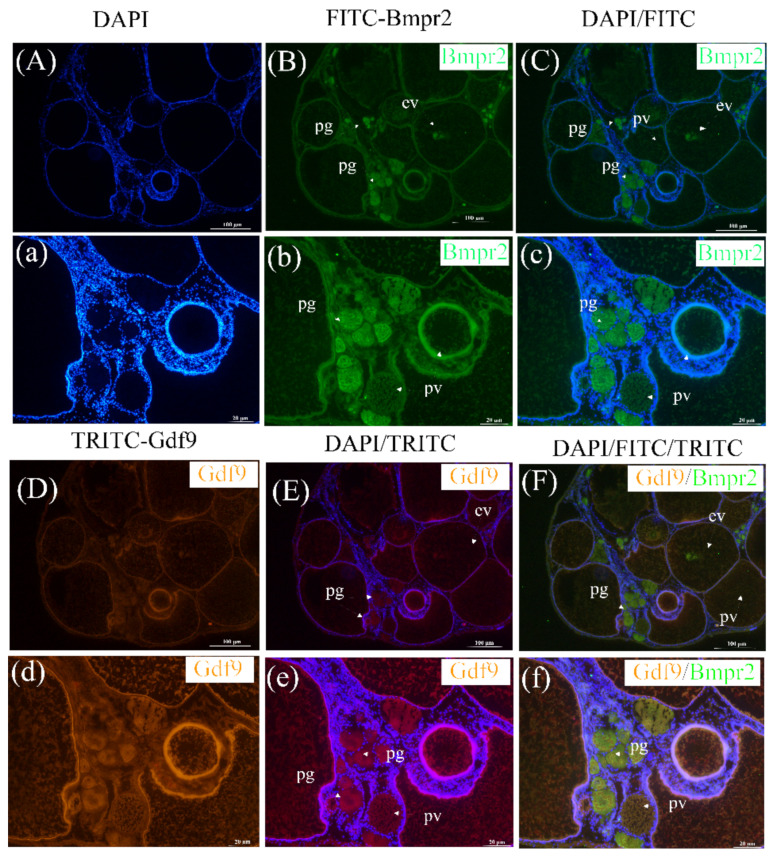
Immunofluorescence analysis of Bmpr2 and Gdf9 in the ovary of ricefield eel, *Monopterus albus*. Fluorescent images of Bmpr2 in the ovary captured by FITC: (**A**,**a**), DAPI; (**B**,**b**), FITC; (**C**), merged fluorescent images of (**A**,**B**). (**c**), Merged fluorescent images of (**a**,**b**). Fluorescent images of Gdf9 in the ovary captured by TRITC: (**D**,**d**), TRITC; (**E**), merged fluorescent images of (**A**,**D**); (**e**), merged fluorescent images of (**a**,**d**); (**F**), merged fluorescent images of (**C**,**E**); (**f**), merged fluorescent images of (**c**,**e**). PG, primary growth stage; PV, previtellogenic stage; EV, early vitellogenic stage. Immunofluorescence (green) shows Bmpr2 expression in the ovary. Immunofluorescence (orange-red signal) shows Gdf9 expression in the ovary. Nuclei are labeled with DAPI (blue). All photomicrographs were taken by an Olympus inverted research microscope.

**Figure 6 ijms-23-15349-f006:**
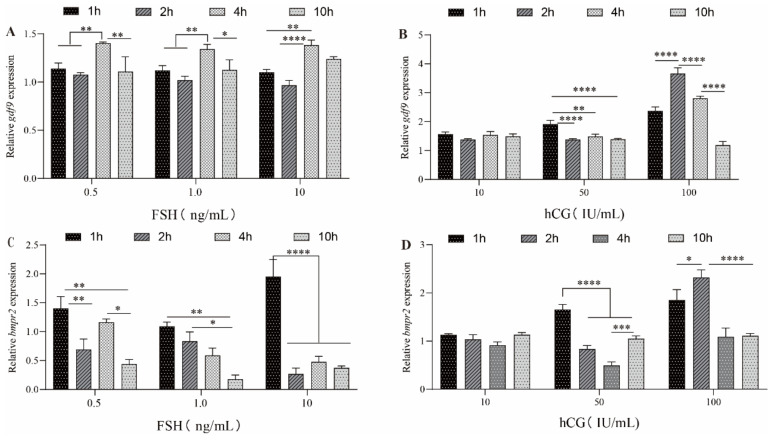
Expression of *gdf9* and *bmpr2* in ovarian tissue of ricefield eel after FSH and hCG incubation in vitro. Results are expressed as means ± SEM (error bars), which were compared with the control group. (**A**,**C**) The relative expression level of *gdf9* and *bmpr2* after FSH incubation; (**B**,**D**), the relative expression level of *gdf9* and *bmpr2* after hCG incubation. FSH, follicle-stimulating hormone; hCG, human chorionic gonadotropin; IU, international unit. *, **, *** and **** are the significant differences in Duncan’s multiple comparisons, representing * *p* < 0.05, ** *p* < 0.01, *** *p* < 0.001 and **** *p* < 0.0001, respectively.

**Table 1 ijms-23-15349-t001:** Primers used for cloning and RT-qPCR.

Primer	Sequences (5′–3′)	Primer	Sequences (5′–3′)
*bmpr2*F1	AAAGAAGGGGACCAGAAGAA	*bmpr2*R1	CATGCGGTAAGCAAAGAAAG
*bmpr2*F2	TGACCAACCTCCCTCCGCAG	*bmpr2*R2	GATTGTCCCCACCTCACTTA
*bmpr2*F3	AAGGGATGGTCTGTGTACGT	*bmpr2*R3	ACGTACACAGACCATCCCTT
*bmpr2*F4	GAGACGATGGAGGACTGTTG	*bmpr2*R4	AGTGGTGGTTGTAGTGGAGG
*bmpr2*F5	TCCAAGAGACACGTCATCGA	*bmpr2*R5	GCCCAACAGACCAGACACAA
*bmpr2*qF1	AGGCAGGGAACCACCCA	*bmpr2*qR1	GGAGCGAACCGCCAAAC
*gdf9*qF1	AGAAGGTGGAGAGGGAATC	*gdf9*qR1	GAAGTCATACAAGGCACATCA
*ef1α*qF1	CGCTGCTGTTTCCTTCGTCC	*ef1α*qR1	TTGCGTTCAATCTTCCATCCC
*rpl17*qF1	GTTGTAGCGACGGAAAGGGAC	*rpl17*qR1	GACTAAATCATGCAAGTCGAGGG

F: sense primer; R: antisense primer.

## Data Availability

The datasets supporting the conclusions of this article are included within the article.

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
