# Peer review of "Expression Patterns and Gonadotropin Regulation of the TGF-β II Receptor (Bmpr2) during Ovarian Development in the Ricefield Eel Monopterus albus"

_ijms, 2022, doi:10.3390/ijms232315349_

Round 1

Reviewer 1 Report (New Reviewer)

In my opinion, the results obtained by the Authors are an important contribution to a deeper understanding of ricefield eel reproduction, which is a commercially important fish species in China. Therefore, I believe that the presented results and undertaken research are justified and can help in understanding the reproductive phenomenon of this fish species.

Minor questions and suggestions:

1. Introduction: What I miss is the one clear paragraph at the end of this section about what was the main aim of the study, not only what the Authors did.

2. Results:

line 129 - why Authors used four different levels of significance?

line 138-139 - the abreviations PG and EV should appear earlier in the text,  when they were first used.

3. Methods:

line 297 - please provide the range of temperature and what photoperiod was used.

line 302 - temperature of liquid nitrogen is definitely lower that -80 degrees of Celcius. I think that Author meant that the samples after collection were preserved in liquid nitrogen and later kept in -80 for RNA extraction? Please specify.

line 304 - please provide the methodology for histological sampling and staining.

line 409 - if p<0.05 was considered as statystically significant why Authors used also p<0.01, p<0.001 and p<0.0001?

4. Conclusions:

What I miss in this section is - what should future studies on this topic look like? What other questions should we ask? What should we do or think differently about ricefield reproduction after this findings? Whwrw could we use this new information?

Author Response

Response to reviewer 1

Dear reviewer,

Thank you very much for reviewing our manuscript. We appreciate your comments, which are very pertinent, constructive and professional. All modifications in the revised manuscript are marked in red color. The followings are our point-by-point responses:

Comment:

1.Introduction: What I miss is the one clear paragraph at the end of this section about what was the main aim of the study, not only what the Authors did. 

Response: The aim being to explore the temporal and spatial distribution of bmpr2 in the ovary of ricefield eel and the effects of FSH and hCG on the expression of bmpr2 and gdf9 in the ovary of ricefield eel. These results will provide a better understanding the functions of bmpr2 and the regulatory mechanism of gdf9 in ovarian development, and be helpful to provide the basic data for the research of artificial propagation technology of ricefield eel. (Lines80-82).

  1. Results: line 129 - why Authors used four different levels of significance?

Response: The data were analyzed using One-way (ANOVA) followed by Duncan’s multiple comparisons test using SPSS 21.0 (IBM, Armonk, NY, USA). Based on the significant differences of Duncan’s multiple comparisons, “*, **, *** and ****” means the significant levels “0.05, 0.01, 0.001 and 0.0001”, respectively.

  1. line 138-139 - the abreviations PG and EV should appear earlier in the text, when they were first used.

Response: Thanks for your suggestions. PG and EV appeared in Lines 137-144 for the first time, and we also explained them. This is the second occurrence in Lines 322-330 in the part of “Materials and methods”.

  1. Methods:

line 297 - please provide the range of temperature and what photoperiod was used.

Response: The range of temperature and photoperiod have been added to the manuscript (Lines 311-314).

  1. line 302 - temperature of liquid nitrogen is definitely lower that -80 degrees of Celcius. I think that Author meant that the samples after collection were preserved in liquid nitrogen and later kept in -80 for RNA extraction? Please specify.

Response: Thanks for your suggestions. The collected tissue samples stored at - 80 ℃ for RNA extraction after quickly freezing with liquid nitrogen. (Lines 319-320).

  1. line 304 - please provide the methodology for histological sampling and staining.

Response: According to the method of He et al [18], the rest part of gonads were used for histological observation after a series of dehydration, embedding, sectioning (the 5 µm sections) and staining with hematoxylin-eosin. (Lines 322-324).

  1. line 409 - if p<0.05 was considered as statystically significant why Authors used also p<0.01, p<0.001 and p<0.0001?

Response: In this study, the data were analyzed using one-way (ANOVA) or two one-way ANOVA followed by Duncan’s multiple comparisons test using SPSS 21.0 (IBM, Armonk, NY, USA). Although “p<0.05” means that there is a significant difference between the two groups of samples, Duncan's multiple comparative analysis results show that there are four different significant levels, which are 0.05, 0.01, 0.001, 0.0001 respectively. Therefore, it is displayed according to Duncan’s multiple comparisons test results.

  1. Conclusions:

What I miss in this section is - what should future studies on this topic look like? What other questions should we ask? What should we do or think differently about ricefield reproduction after this findings? Whwrw could we use this new information?

Response: we add some information. Our results showed that FSH and hCG can affect the expression of gdf9 and bmpr2 in the ovary of ricefield eel, however, the regulatory mechanism remains to be explored. Whether FSH and hCG helps in artificial breeding needed to further research (Lines 451-458).

Reviewer 2 Report (New Reviewer)

Very interesting and high quality MS. I believe it should be accepted as corrected. My detailed comments are included in the text. To see them all, open the file in Acrobat Reader.

Author Response

Response to reviewer 2

Dear reviewer,

Thank you very much for reviewing our manuscript. We appreciate your comments, which are very pertinent, constructive and professional. All modifications in the revised manuscript are marked in red color. The followings are our point-by-point responses:

  1. why use use FSH and hCG - it is unclear for the readers. FSH it is "natural" hormone but hCG is not specific in fish. Of course both of them are used in artificial maturation of many finfishes (FSH as a part of pituitary gland homogenate) - e.g. Animal Reproduction Science, 2021, 225, 106684; Animal Reproduction Science, 2020, 221, 106543 as in some eel spcies, e.g. Aquaculture International, 2015, 23(1), pp. 13–27

Response: Thanks for your suggestions. FSH and hCG are gonadotropins that can regulate ovarian development (doi:10.1016/0016-6480(73)90078-6). FSH comes from the pituitary gland and is part of the pituitary homogenate. they are used in artificial maturation of many fishes. For example, Anguilla anguilla (doi:10.1007/s10499-014-9794-2), Gymnocephalus cernua (doi:10.1016/j.anireprosci.2020.106684) and ide Leuciscus idus (10.1016/j.anireprosci.2020.106543). It has been added in the Introduction part (Lines 77-80).

  1. Did you use any anaesthetics? If yes, add drug name and dose. If not - describe (or add reference) how you killed the fish

Response: Fish were decapitated after anaesthetized with MS-222 (100 μg/mL, Syndel, Washington, USA) for 10 minutes. Then, the gonads, pituitary, heart, liver, kidney, intestine, spleen, blood and other tissues of ricefield eels were collected and and then stored at - 80 ℃ for RNA extraction after quickly freezing with liquid nitrogen. (Lines 319-321).

  1. but the minimum DO should be recognized. If you applied any kind of areation or oxygenation of the water, please add this information. The oxygen level, especially low, might influence of the obtained results

Response: Ricefield eels were cultivated at condition of temperature of 21.7±2.5℃ and photoperiod of 16 h light:8 h dark. The fish were temporarily maintained in the laboratory under a natural condition. Various indicators of aquaculture water environment had been be taken into consideration. The dissolved oxygen is more than 6 mg/L, pH is 6.5-7.4, and ammonia nitrogen is less than 0.2 mg/L (Lines 311-314).

Round 2

Reviewer 2 Report (New Reviewer)

The MS was corrected. Excellent work

This manuscript is a resubmission of an earlier submission. The following is a list of the peer review reports and author responses from that submission.

Round 1

Reviewer 1 Report

The manuscript ijms-1932795 could be interesting, but has several flacks from a methodological and an analytical point of view that have to be amended before being suitable for publication.

In addition, English had to be check in deep, as in some part of the manuscript is very difficult to understand the ideas expressed.

As general concern, authors should provide enough data that support that the antibodies used specifically bind the proteins that they are supposing. It could be determining that the epitope of those antibodies is well conserved through evolution or at least in the mammalian species in which the antibodies are produced and M. albus. If not they had to demonstrated with data, that the antibodies specifically react with the M. albus specific proteins. This issue is important taking into account that the IHC micrographs that authors show are not clear and the staining is not very strong. The pattern of the staining observed in the micrograph is quite similar to an unspecific staining.

Specific comments:

Line 40-41, please check the English use in this sentence.

Line 42-44. Are this sentence reference what happened in mammals? Please add, if it is the case.

The second and third paragraph of the introduction should be only one.

Line 68, The target gene? I guest that authors might say target cell, is not it? Normally a molecule induce changes in a cell through a specific receptor signalling?

Line 69, A reference about this stayment is needed at this point.

Line 72, what the authors had incubated with hCG? Normally you incubate a tissue or a cell line with an hormone or a molecule. This sentence has no sense. Please re-write it.

Line 85-87, what is referring this sentence to? I guessed that this sentence is comparing the Bmpr2 cloned at this work with the Bmpr2 of the two species specified in the sentence. Please clarify this issue and rewrite the sentence. 

Fig 1. The stop codon highlighted in red is not TAA u TGA. What the asterisk underneath represent? Please amend.

Figure 2, legend. A revision of the Englihs structure of the sentences is needed. Moreover, y ou have 7 different species in the figure with the exception of the one that you clone. However, you only report the accession numbers of 6. Please check and amend this issue.

Line 113, check English again.

Fig. 4. The load control of the western blot is missing. Author should show the actin-b of both Bmpr2 and Control as control of load. Why the authors did not included the other tissues in the western blot analysis. Pituitary for example  or testis also. What the control means? Was a western blot of actin-b or what is it?

Lines 1256-130, It is the first time in which the abbreviations related to the ovarian cycle appeared in the text. Authors should included what they means, here.

Fig. 5, I cannot see reaction in a and b micrograph. It could probably be unspecific reaction. The slightly more intense staining in the OG could be specific reaction, but not in the PG. In order to clarify whether the specific or unspecific reaction levels of the IHC staining, authors should provided negative and preadsorved controls. 

These controls are very important taking into account that fish ovary should have a quite intense unspecific reaction with a lot of antibodies.

Fig. 6, I am not sure what I am looking at. Could the authors point to the ovocytes somehow?

It seems like now the Bmpr2 is appearing in more oocytes than in the last figure. How could be possible?

The magnification is not clear enough and it is difficult to asses is we are looking at the ovary or inside one oocytes. Please could the authors include a scale bar in this pictures as they did in the last figure.

Line 152, This point needs further analysis and a deep revision in order to accurate the description to the data obtained and the statistical analysis performed.

Line 155, I cannot see this trend, taking into account the error of the graph, there are not differences between 1 and 10 ng/mL response. There is a slight up-regulation at 4 hours in the 10 ng/mL doses compared to 2 h of estimulation but not comparing with other times.

Line 156, The description is not right. Authors statistically compared between times within the same dos exposured group but not between exposure groups at the same time. In fact, there are not differences between exposure groups at certains times at other it could be, but authors should perform and show the statistical analysis in order to determine the veracity of this sentences.

Line 161, compared with what situation?

Line 172, How many sequences did the authors cloned? Please check the English language

Line 181, this sentence is very difficult to understand. Is at least one of this two transmembrane domains highly conserverd or not or what is different in the M. albus Bmpr2 comparing with other Bmpr2 of other species.

Linew 184, which are the classical structural features of..... Authors should explain this and included references.

Line 187, this sentence has no sense. What are the main differences between Bmpr2 of M. albus expression and other fish species.

Line 217, Expression or protein production ?

Lines 220-221, difficult to understand, this sentence has no sense.

Line 227, in which animal, what this study about?.

Line 239, the authors did an in vitro experiment of 10hours maximum. Authors cannot conclude anything about ovarian development in 10 hours. Please rewrite this sentence to adjust it to the data obtained. the authors observed a slight up-regulation when comparing 2h with 4h of exposure but not when comparing 1h with 4 h of exposure, so the obvious upwards, is not so obvious.

In order to demonstrated that Gdf9 is involved in ovarian development, authors need to treat with Gdf9 and demonstrated that the ovary really go further in its developed status.  The data showed to demonstrated the relation between FSH and Gdf9 are not so consistent as authors wished.

Lines 256-257, authors did not compared between different doses within the same time, so I do not understand this part of the sentence. It is not supported by the data.

Line 335, Authors must specify all the data about the primary antibody. Was this antibody a commercial one? Was specific for M. albus Bmrp2 or was specific for other species? In that case, did the author anything to demonstrate its specific binding to M. albus Bmrp2.

Lines 346, As the primary antibody is specific for rabbit Bmrp2, authors need to included the data that demonstrated that the epitope of the antibody is well conserved between both species or any preadsorved control that demonstrate that even when the epitope is not conserved, the antibody is able to specifically bind M. albus Bmrp2 and not other proteins in the samples.

This is very important because the staining observed in the micrograph of the manuscript is not very intense and  could be easily misunderstanding with unspecific staining.

Line 358, similarly preadsorved and negative controls are needed. Authors need to demonstrate first that the antibodies specifically interact with M. albus Bmpr2 and Gdf9 proteins and not with other.

Author Response

Dear reviewer,

Thank you very much for reviewing our manuscript. We appreciate your comments, which are very pertinent, constructive and professional. All modifications in the revised manuscript are marked in red color. our point-by-point responses has been uploaded.

Reviewer 2 Report

For He et al., on BMPR2

Line 56  - in ovarian – in the ovary

Line 62 – significantly

Line 67 – typo

Line 68 – transferred?

Line 69 – effect – role

Line 71 = evaluated mRNA and protein levels == This is only partly true. Protein levels were not determined at all!

Line 81 – N-glycation – N-glycosylation

Line 85-87 – This sentence starting with “Only… “ is unclear.

From Line 88 onwards, I am not pointing out the typographical errors anymore. English editing is not a referee’s responsibility.

Line 97 – Figure 2 does not show a three-dimensional structure of Bmpr2.

Line 99 – The statement “Conserved amino acid sequences are indicated with different colours.” is incorrect. Usually, conserved amino acid sequences are indicated by the same colour. Also, there are non-conserved residues that are represented by the same colour (between 503-600 residues, 689-781).

Line 115 – were obviously – significantly

Line 119 (Figure 4) – The authors should show the full Western blot photo as a Supplementary Figure. The photo should show the size ladder. The authors should also define the Western blot ‘Control’. Figure 4B in the present form is meaningless.

Lines 123-124, 137-138, 166-167 – What do those asterisks represent exactly? In the statement beginning with “Numbers represent…”, which numbers are being referred to as Pearson correlation coefficients? The statistical analysis described in Section 4.9 described ANOVA and DMRT as the only analytical methods used when data were compared.

Line 133 – The oocyte shown in Figure 5c and enlarged in 5d is already vitellogenic.

Discussion

The first paragraph of the Discussion, and the entire Discussion section anyway, contains contradictory and vague statements. On the third paragraph, the terms used to describe ovarian/oocyte stages are imprecise. The term ovary/ovaries is loosely used to refer to the location of bmpr2 expression but in the results (Lines 141-142) the authors claim bmpr2 signal was clearly detected in the oocyte cytoplasm (Figure 6B), as is also stated in Line 224-225. It is not clear what the justification was of testing the effect of hCG in immature ovarian tissues.

The results of the study altogether do not support the proposed model shown in Figure 8. It was never stated in the Methods that the Bmpr2 and gdf9 were examined in ovarian stages from follicular growth to oocyte maturation.

Methods – It is not sufficient to just cite a reference for a particular method as the previous studies were done on different species.

Line 288-289  – ovaries parts – ovarian tissues of ovarian fragments

Line 328 – For the Western blot analysis, the primary antibody used was designed for human Bmpr2. Was there a previous validation that it can be used in ricefield eel?

Line 331-333 – Please add a line or two how the proteins were extracted from the ovaries and how the Bmpr2 proteins were isolated before transferring to the PVDF membrane. What is meant by discontinuous gel electrophoresis?

Line 334, 337, 345 – hatched – incubated or blocked

Line 335 – Was there a particular reason why the 5% non-fat milk was prepared in 10 mM PHOSPHORIC ACID buffer?

Lines 342 to 367 – Sections 4.6 and 4.7 – What were the negative controls for these analyses? What was/were the ovarian stage/s used for these experiments? For the immunohistochemical and colocalization analysis, what were the sources of Bmpr2 and GDF9 antiserum? Were the tissues serially sectioned?

Line 368 – Section 4.8. Was any antibiotic added to the culture media (normal saline)? The method states that bmpr2 expression was analyzed by RT-qPCR, however Figure 7a and 7c show gdf9 relative expression levels. There are no gdf9 primers included in Table 1.

Line 369 – freshly – fresh

Lines 369 – 373 – Section 4.8 – What were the sources of hCG and FSH? For FSH, from what species does it correspond?

Lines 374 – 380 Section 4.9 – Statistical analysis – This section does not say a correlation analysis was performed which contrasts with what are being mentioned in Lines 123-124, 137-138, 166-167.

Line 375 – typo

Line 377 typo

Lines 384-387 – The data presented are insufficient to support these conclusions. In the absence of negative controls and explanation regarding the sources of the antibodies used, the immunohistochemistry and colocalization results are not valid.   

Author Response

(The authors gave the same response as above.)

Reviewer 3 Report

The authors have carried out a series of methods aimed at a better understanding of the function of the bmpr2 receptor in the ovarian development of the ricefield eel, Monopterus albus. However, I have several major concerns, in particular, concerning immunohistochemistry. 

Initially, I would like to query the authors why there are no control figures for immunohistochemistry analysis, as well as any reference to its performance. The authors should add immunofluorescence controls in both immunohistochemistry approaches. It is important to include these negative controls in the figures referred in the text.

In the figure 5, the authors said that although Bmr2 receptor had been expressed in oocytes at all developmental stages, immunohistochemistry analysis did not detect a positive signal, at least in the theca cells and GCs. However, we can observe several areas immunostained in the image that even the authors have highlighted. I was wondering, what does it mean? Does it a background noise? In addition, I missed more information about different parts of the ovary in the picture. The authors should take more care in making their writing and figures to give the lectors as much possible information.  

Throughout the text, we can read several references about the localization of the Bmr2 receptor and Gdf9, but it does not clear which group of cells are localised in this manuscript. Hence, the authors have performed immunofluorescence against Bmpr2 and Gdf9 in the ovary. However, in the figure 6 is hard to differentiate the vesicle stages, and there is not enough information in the images and the results section. Those pictures are not clear, and I am not sure about the positive results that the authors are reporting here. The authors said that the immunostaining is detected in the oocyte cytoplasm but that affirmation is not significant visible. In the fluorescent images of figure 6, the background noise makes it quite hard to differentiate what structure/or cell layer we are seeing. In addition, there are some structures/vesicles that also are immunostained in all images. How could the readers differentiate which cells are positive immunostained or not if there is no indication of it? It is necessary to make the negative controls to confirm the specificity of the immunostaining. Furthermore, I did not find information about the antibodies used in the text. Likewise, to improve the interpretation of the results, the authors should also complement the images with representative images of the haematoxylin-eosin staining section. These new images will allow potential readers to identify the different cell layers in the ovary. In addition, I recommend including some photos with more magnification and the scale bars of the images (there are not scale bars in the figure 6).

On the other hand, several studies have reported that lipofuscin, granules accumulate in the cytoplasm of LAF cells as a result of lysosomal digestion of the oocyte components, could have autofluorescence in different. It would be interesting to check this before the immunofluorescent analysis. In addition, the high concentration of antibodies (1:50 for the primary antibodies and 1:50 for the secondary antibodies) used in this study could mask the results. 

In “in vitro” analysis, freshly ovarian fragments were divided into three different groups, consisting of two experimental and one control group. The experimental group were treated with Fsh and hCG with three different doses and time lags, while the control group was incubated with normal saline. However, in the figure 7 is not represented the control group that is important for the analysis. Conversely, the authors have shown a significant relative expression between times for one dose that does not provide relevant information. With this experimental design, I guess that the authors want to know whether the Fsh and hCG treatments have some effect on RNA expression levels of bmpr2 and gdf9, and that is why it is relevant to include the control group. The author should repeat the statistical analysis including this last group. On the other hand, the data analysis used in this manuscript was performed by one-way ANOVA. However, it is not correct to use two one-way ANOVA in the same statistical analysis when we are studying the effect of two different factors, such as time (1, 2, 3, 4, 10 hours) and treatments (Control and experimental groups). The analysis must be repeated by using two-way ANOVA.

Line 58. In zebrafish, two different Bmp receptors have been found (bmpr2a and bmpr2b). I was wondering whether the ricefield eel species has also two receptors. Do the authors know what kind of receptor was cloned, bmpr2a or bmpr2b? If there are other isoforms it would be interesting to include different isoforms of this receptor in the phylogenetic analysis.

Line 187. What is the point of the following sentence? I do not understand it. “Although the expression level of bmpr2 seemed to vary among different tissues, it was found to be expressed in different tissued examined”

Line 159. Fig. 7, remove Fig. 6

Line 291-296. Without DNase treatment, I am sure that your samples are contaminated with gDNA. This contamination could invalidate the results for the real-time qPCR. Was there prior digestion with DNase before cDNA synthesis? If so, please reference it in the text. 

Author Response

(The authors gave the same response as above.)

Round 2

Reviewer 1 Report

Authors had improve their manuscript. However they shoudl included the information relate to the specificity of the antibodies used, somewhere in the material and methods section in order to give suuport to their data. 

Authors send this information in the letter to the reviewer, but they did not included in the manuscript.

BMPR2 (LifeSpan BioSciences, Seattle, WA, USA; cat. no. LS‐C178875) was purchased as primary antibody. The primary antibody used was designed for human Bmpr2. sequences alignment of Monopterus albus Bmpr2 with sequences of Homo sapiens BMPR2 (LS‐C178875) was performed by the DNAMAN 6.0 software translation tool. The homology of these two sequences was 82.17%. The pre-experiment also verified that BMPR2 (LS-C178875) can be used for Monopterus albus.

Reviewer 3 Report

In vitro analysis, fig 6.

I am still having questions about what the results of the figure 6 mean. The authors must compare the different treatment effects with the control, but the results in the figure 6 still show significant relative expression between times for the same treatments but not with control. The results of the figure 6 do not reveal what the text said. I am not able to see any effect of the treatments compared to the control groups. On the other hand, I have observed that the error bar has changed concerning the old figure 6 (it was before the figure 7), Could the authors explain that phenomenon? If there are any changes about how they made the figure the authors should say it.

I do not understand why the authors have eliminated the Immunohistochemical analysis. 

In the phylogenetic analysis.

The authors said that there are not two isoforms of bmpr2 in Monopterus albus and I guess for that reason they have not wanted to make a new phylogenetic analysis including others bmpr2 isoforms, but still think that including them in the phylogenetic analysis would be interesting.